# Estimate of Anemia with New Non-Invasive Systems—A Moment of Reflection

**Giovanni Dimauro** [1],* , **Serena De Ruvo** [1], **Federica Di Terlizzi** [1], **Angelo Ruggieri** [1], **Vincenzo Volpe** [1], **Lucio Colizzi** [1] **and Francesco Girardi** [2]

1   Department of Computer Science, University of Bari, 70125 Bari, Italy; serederuvo@gmail.com (S.D.R.); federica.diterlizzi1@gmail.com (F.D.T.); angeloruggieridj@gmail.com (A.R.); vincenzovolpe4897@gmail.com (V.V.); lucio.colizzi@gmail.com (L.C.)
2   UVARP Azienda Sanitaria Locale, 70132 Bari, Italy; francesco.girardi@asl.bari.it
*   Correspondence: giovanni.dimauro@uniba.it

**Abstract:** Anemia is a global public health problem with major consequences for human health. About a quarter of the world population shows a hemoglobin concentration that is below the recommended thresholds. Non-invasive methods for monitoring and identifying potential risk of anemia and smartphone-based devices to perform this task are promising in addressing this pathology. We have considered some well-known studies carried out on this topic since the main purpose of this work was not to produce a review. The first group of papers describes the approaches for the clinical evaluation of anemia focused on different human exposed tissues, while we used a second group to overview some technologies, basic methods, and principles of operation of some devices and highlight some technical problems. Results extracted from the second group of papers examined were aggregated in two comparison tables. A growing interest in this topic is demonstrated by the increasing number of papers published recently. We believe we have identified several critical issues in the published studies, including those published by us. Just as an example, in many papers the dataset used is not described. With this paper we wish to open a discussion on these issues. Few papers have been sufficient to highlight differences in the experimental conditions and this makes the comparison of the results difficult. Differences are also found in the identification of the regions of interest in the tissue, descriptions of the datasets, and other boundary conditions. These critical issues are discussed together with open problems and common mistakes that probably we are making. We propose the definition of a road-map and a common agenda for research on this topic. In this sense, we want to highlight here some issues that seem worthy of common discussion and the subject of synergistic agreements. This paper, and in particular, the discussion could be the starting point for an open debate about the dissemination of our experiments and pave the way for further updates and improvements of what we have outlined.

**Keywords:** anemia; hemoglobin; human tissues; conjunctiva; non-invasive medical device; image analysis

---

## 1. Introduction

Anemia is a global health problem: the World Health Organization estimates that a quarter of the world's population has to deal with anemia problems. Anemia is defined as the decrease in red cells in the blood or as a decrease in the concentration of hemoglobin (Hb): consequently, it reduces the ability of the blood to transport oxygen. It is mainly caused by nutritional factors, infectious diseases or genetic factors. Severe anemia can compromise the availability of oxygen supplied to the cells and cause damage to vital organs [1–6]. Iron deficiency anemia is the most

common nutritional deficiency, and it causes thousands of deaths; regretfully, it is also responsible for increased morbidity and mortality in pre-school children and pregnant women. Since 2002, iron deficiency anemia was considered to be among the most important contributing factors to the global burden of disease [7–10]. Furthermore, a large population survey showed that nearly ten percent of the elderly, without distinguishing between genders, were anemic. Mild anemia cases represent the majority, having estimated a summary percentage of 4% for all other grades. The percentage of male anemic patients reaches the minimum between 17 and 49 years. For women, the condition occurs between the ages of 50 and 64 years [11]. One-third of anemia cases in adults are attributable to iron deficiencies, folate and vitamin B12. Iron deficiency anemia summarizes approximately 50% of nutrient-scarred anemia cases, of which bleeding caused by gastrointestinal lesions is the first cause.

In many cases, anemic patients need transfusions based on their hemoglobin level, and therefore, need to be monitored frequently. It has a slow evolution, in fact, there are no obvious symptoms until the concentration of hemoglobin in the blood becomes really low, because the human body activates compensatory processes. When compensation can no longer guarantee an adequate dose of circulating oxygen, several symptoms appear. Although the symptomatology may vary according to the degree and type of anemia, distinctive features often accompany its presence. Recurrent symptoms common to all types of anemia include fatigue, dizziness or light-headedness, headache, pallor, chest pain, weakness, irregular heartbeat, shortness of breath, and cold hands and feet.

To carry out a correct diagnosis, careful study of the anamnesis, an effective physical examination and a series of essential instrumental examinations are necessary. During the physical examination, heart rate, respiratory rate and conjunctival pallor of the tongue or nail bed are evaluated in [12–19]. The physical examination, therefore, assumes an important role in the diagnosis process, and whatever innovative tool is designed, it must take this into account. The last step is blood cell counting that will provide answers to expectations: normally, the hemoglobin level is measured invasively. Often the use of invasive methods is not recommended, for example in the case of infants, the elderly, pregnant women, patients with anemia and sickle cell. In addition, frequent blood sampling creates significant discomfort to the patient and is quite expensive, especially in areas of the world that have limited economic resources. For this reason, it is of great interest to study methods and design tools that allow us to monitor the hemoglobin concentration in a non-invasive way, with reduced costs, both in the laboratory and at home, sometimes even daily.

Many studies [13,19–21] show interest in the pallor of the exposed tissues of the human body to estimate anemia. Pallor is characterized by a lack of color in the skin and mucous membranes due to a low level of circulating hemoglobin. This may be evident on the entire body, but is easily observed in areas where blood vessels are close to the surface, such as the palm, the nail bed and mucous membranes such as the tongue or conjunctivae.

There are numerous non-invasive methods and tools that indirectly measure the value of hemoglobin in the blood and the level of oxygen in human tissues. These include techniques such as photoplethysmography, reflectance spectroscopy and fluorescence spectroscopy of oral tissue, but many of them are not available at affordable costs and are often not available as portable or wearable technologies.

The aim of this paper is to provide a commentary to highlight some issues that seem worthy of common discussion and the subject of synergistic agreements. Specifically, we discuss the economic and social implications related to these technologies. Further on, focusing on the more scientific aspects, we highlight some clues about the exposed human tissues that are subject to clinical analysis as a privileged region to estimate anemia and propose a brief overview of the smartphone-based systems to support an effective diagnosis of anemia, including their technical limitations. Then important differences in carrying out the experiments and in presenting the results of research works on this topic are presented and finally discussed.

## 2. Technologies to Power the Healthcare and Medical Service in Estimating Anemia

Hence, we wonder if personal, low cost and non-invasive medical devices, eventually supported by artificial intelligence, can power the healthcare and medical service to discover and treat anemic patients. It is extremely difficult to aggregate the reports of different international organizations and national institutions and it is, therefore, impossible to create a dashboard to provide a public update on the spread of the disease, with data acquired from validated sources. Surely this pathology is present around the world, even in more economically advanced areas, and the available estimates suggest a phenomenon of great impact on the world population also if with marked differences in the local dimension of the epidemic. Consequently, it is impossible to estimate the cost of coping with the disease: prevention (or for failure to prevent), blood analysis, instrumentation, personnel and facilities [12].

It is arduous to estimate the social cost for anemic patients or those suffering from other pathologies that are also anemic, who must frequently undergo laboratory analysis. Well, the scientific evidence in the literature and the results described in this paper show that digital technologies can make it possible to exclude from the laboratory analysis a good part of the population that is certainly not affected by this disease. Furthermore, they can provide a reliable alarm on people with suspected severe anemia.

Like other diagnostic-clinical and analytical-laboratory medical disciplines that begin to make extensive use of the image, sound or signal analysis, machine and deep learning techniques [22–27], it is worthwhile to invest in research and development of technologies such as those presented in this paper, with the dual purpose of significantly reducing the costs borne by the national health systems and power the healthcare and medical services that would be exempted from a considerable amount of practically useless activities.

The instrumentation, diagnostic processes and therapeutic applications have undergone considerable technological improvement and also advances in digital image processing are playing a key role in the medical field gaining an important role also in the physical examination in the diagnostic process. While minimizing the subjective nature of an observation test falls within the primary objectives, machine learning systems are winning solutions, using statistical methods for the recognition of patterns that are difficult to generalize by human beings.

Expanding these current research and the development of new technology may significantly reduce the costs borne by the national health systems, power the healthcare and medical services that would be exempted from a considerable amount of practically useless activities and finally (but not in order of importance) to obtain positive effects on patients, for example, those who must frequently undergo laboratory analysis: this is a value that is difficult to estimate but even more relevant than the previous ones.

In the next two sections, we explore the clinical evaluation of anemia oriented to the analysis of different exposed tissues and the most promising efforts dedicated to the measurement of hemoglobin by analyzing the color of the fingertips, nail bed and the palpebral conjunctiva.

## 3. Clinical Evaluation of the Pallor of Exposed Human Tissues

The validity of the diagnosis of anemia through the clinical evaluation of the pallor of the exposed human tissues is evidenced by numerous findings in the literature, some of which are reported below. It must be said that the authors explain that many factors can influence the reliability of this evaluation, for example the severity and causes of anemia, the different pigmentation of the skin that can influence the interpretation of pallor, for example in the palm of the hand.

To diagnose severe anemia, as described in [17], IMCI (Integrated Management of Childhood Illness) suggests using the paleness of the palm, but since a greater presence of melanin can influence the color of the palm, the authors suggest to add further signs examination: they show that it is possible to correctly classify the anemia status of many children, through simple clinical signs, for example conjunctival pallor, which can therefore be added to the guidelines for estimating anemia in children.

Da Silva, et al. [19], compare pallor with the level of hemoglobin in the blood and evaluate the correlation between the opinions expressed by different doctors. The authors point out that the ideal condition for examining skin color is with natural daylight or without direct light on the skin. They demonstrate that the pallor of the conjunctiva is the most accurate in the case of beta-thalassemia, regardless of age and gender and that the pallor of the palm is less reliable for a clinical evaluation, compared to the pallor of the conjunctiva.

In Aggarwal, et al. [28] it is highlighted that the sensitivity of the palmar pallor is low and the specificity is moderate based on different levels of hemoglobin. The authors highlight some limitations, such as the color of the skin which varies depending on the country and also on the cleaning of the hands. In addition, in Spinelli et al. [16] the authors find a greater sensitivity for the pallor of the conjunctiva rather than for the pallor of the palm.

Further, Sheth et al. [18], show that conjunctival pallor can be a more accurate factor in the presence/absence of anemia than palm or nail bed pallor. In the same study, cases were reported in which patients with severe anemia did not always have a pale conjunctiva emphasizing that inflammation in the conjunctiva could cause a false red color.

Kalantri, et al. [29] argue that an accentuated paleness of the tongue could be used as a screening test for detecting the hemoglobin level, with great advantages: it is inexpensive, painless and can be performed in a large population in a very short time. The authors argue that the pink tongue can reassure doctors that a hemoglobin test is not needed; however, it seems that this test cannot be performed and generalized even for children.

Pallor, as reported in Yalçın [20], is useful in evaluating patients with suspected anemia, but can be misleading in the case of high concentration of ferritin, melanin and bilirubin. Furthermore, they show that paleness of the palm has high sensitivity and low specificity for the detection of anemia in cases of beta-thalassemia.

## 4. Digital Analysis of Images of Exposed Human Tissues Based on the Use of Smartphones

Actually dozens of articles have been written on the topics discussed in Section 3 and in this section. Since we are not going to write a review, we have considered some articles that we know well and use them to dissect the problems and critical issues that we intend to discuss. Then this is not a selection of works based on quality or importance, but few works that in our opinion have been sufficient to identify important critical issues, on which we want to reason.

Non-invasive approaches for estimating anemia or hemoglobin concentration level are considered to be extremely important, particularly for patients who need frequent blood tests or who have difficulty going frequently to test labs. More generally, this type of analysis is suitable in the increasingly frequent cases of managing patient diseases directly in their homes, through appropriate diagnostic and therapeutic care pathways and services like medical records [30,31].

In the first studies presented in the literature on this topic, the use of digital cameras was proposed for the acquisition of images of human tissues to be examined. With the installation of more performing photographic sensors in smartphones, the latter has replaced the use of digital cameras. In this way, all the proposed methodologies become accessible to anyone, making the screening activity also a low-cost process. Unfortunately, the diversity of photographic sensors installed in smartphones poses problems in the standardization of methods and in the development of universal tools for image acquisition and requires the setting of parameters.

Another problem to highlight concerns the focus; unclear images would lose most of the details, making it difficult, for example, to identify blood vessels when needed. To overcome this drawback, some authors have designed and manufactured accessories for taking photos at close range, ensuring uniformity of the acquired images [1]. The acquisition of images of specific tissues, such as the fingertip, instead requires direct contact with the tissue itself, possibly with the internal flash enabled.

Therefore, the current interest of the researchers has now totally oriented towards smartphone-based systems, with the aim of making the screening activities quick and, as we said,

at affordable costs: these devices could offer the possibility to perform tests in a simple way and immediate, both at home and in the clinic, without requiring the patient to purchase specialized instruments that fulfill this task.

The contribution of image analysis has made it possible to develop promising diagnosis and estimation methodologies even in the case of anemia. In many cases, however, the acquisition, segmentation and processing of complex algorithms remain separate tasks, with the last two often carried out remotely, especially if they require the application of complex algorithms.

Based on the recent state of the art research, three main groups of non-invasive smartphone-based procedures have been identified for estimating anemia or the level of hemoglobin concentration. A first classification of recent studies is certainly based on the type of tissue that is taken into consideration, listed here:

1. Conjunctiva analysis: methods and devices that evaluate colors in digital images of the eyelid or forniceal conjunctiva or both;
2. Fingertip analysis: methods and devices that use digital fingertip videos, for example placed in contact with the smartphone camera;
3. Nail analysis: detection of the pallor of digital images of the nail bed.

What unites the three groups listed above is the fact that the conjunctiva, the fingertip and the nail bed are exposed tissues, highly vascularized and not influenced or little influenced by the presence of melanin. The latter could lead to changes in assessments not only among different ethnic groups.

## 4.1. Estimation of Anemia by Analyzing the Eyelid Conjunctiva

In [32], a personal digital assistant is used to acquire the conjunctiva image and calculate the associated Hb value. Here, the variation in ambient light that affects the acquired image is taken into account and therefore a standard gray card is used, in order to balance the brightness. The images are then transferred to a computer to be processed using algorithms developed by the National Institute of Health Image.

In [33], a device is presented to eliminate the influence of external light sources on the acquired images. It is a common viewer with internal lighting, adapted to acquire images of the conjunctiva and reduce the alteration on the color of the conjunctiva due to ambient light (see Figure 1). To study the correlation between the hemoglobin level and the color space features, the RGB values are mapped with the corresponding values of the CIE lab color model. Both the bulbar and the eyelid conjunctiva are considered, both segmented manually. Finally, through the use of a support vector machine (SVM) classifier, the classification between anemic and non-anemic subjects was studied.

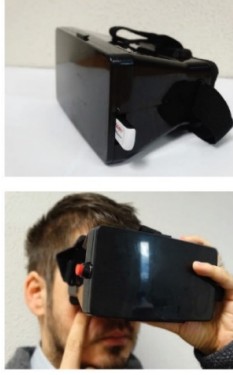

**Figure 1.** Device for acquiring the conjunctiva image.

In [34], the eye region is automatically segmented through the iterative use of the Viola-Jones algorithm. This approach involves a process of improving the image quality and the application

of algorithms that extract the area of interest. Color features are therefore considered to classify non-anemic, mild anemic and severe anemic patients.

In [35], the calibration of digital images is perfected to reduce the impact of ambient light on the area to be analyzed (eyelid conjunctiva). This is accomplished by running two algorithms, the first one based on Mahalanobis distance, and a second one, more robust, which uses a texture-based feature and two color-based features. The robust algorithm bases the classification on a support vector machine and on artificial neural network. Two further features are used to correlate the digital image and hemoglobin level, the high hue ratio and the pixel value in the middle. The same authors considerably reduce the residual indecision of previous studies through the use of the Kalman filter as explained in [36].

In [37], a new tool was proposed for the acquisition of the conjunctiva images (see Figure 2), which solves some problems of the tool presented in [33].

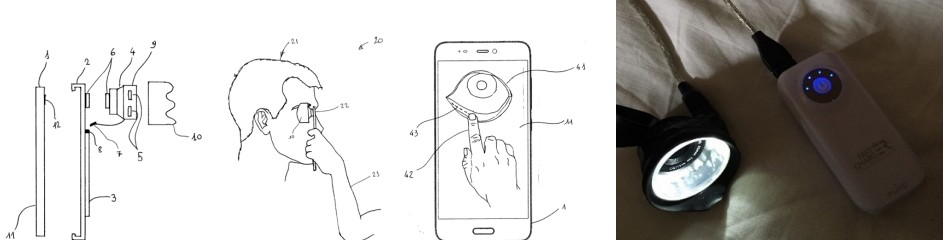

**Figure 2.** Acquisition tool and process to exclude ambient light.

The new device consists of a special spacer and a macro lens to acquire images with a high-resolution smartphone at close range: with this device, images not affected by ambient light are obtained (see Figure 3a). The authors also present a special software, called Hbmeter (see Figure 3b) which allows the acquisition of eye images, the assisted selection of the conjunctiva region (see Figure 3c) and the estimate of the condition of anemia. It is designed to store patient data, which can then be consulted or sent to the doctor. The color of the conjunctiva is assessed using the components of the CIE Lab color space.

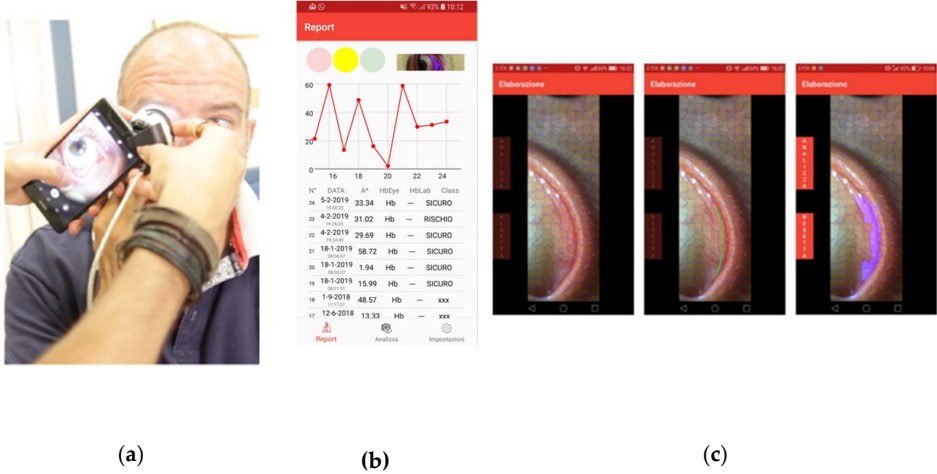

(**a**)  (**b**)  (**c**)

**Figure 3.** (**a**) Image acquisition, (**b**) Hbmeter software, (**c**) assisted segmentation of the eyelid conjunctiva.

In [37], the segmentation of the conjunctiva is obtained manually with the assistance of an algorithm based on SLIC Superpixel, while in [38] the same authors propose automatic segmentation and optimization of the region of interest of the eyelid conjunctiva (see Figure 4). Segmentation of the conjunctiva's area of interest is obtained in two phases: in the first one, image processing algorithms are applied; the second one is based on the study of the brightness of the image.

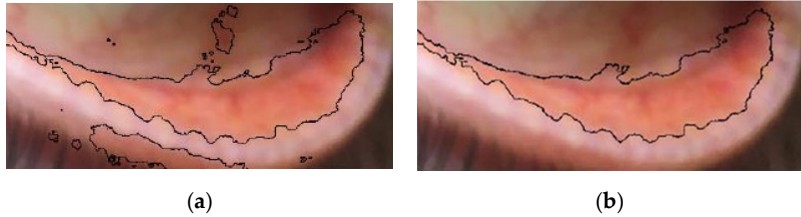

(**a**)                (**b**)

**Figure 4.** Contour selection (**a**) without refining, (**b**) with refining.

In [39], a comparison is made between images taken under ambient light conditions, using a digital camera and a smartphone camera. The authors use a method of standardizing the color of the image with a color calibration card, however, they emphasize that the effect of ambient lighting is not completely eliminated. In any case, they suggest acquiring the images in RAW format to avoid the effect of filters automatically applied by the software of the acquisition device. In this sense, useful experiments should detect the quality and the difference of tissue images also through techniques such as those in [40,41].

In [42], a study was presented that has the ambitious goal of providing health resources to developing countries, where it is difficult to diagnose anemia with traditional methods; in [43] the estimate of the concentration of hemoglobin in the blood is discussed to limit the problem of postpartum hemorrhage caused by anemia, which occurs mostly in Africa and Asia. Both studies are based on the analysis of the color of the conjunctiva, in the second case taking the photos in ambient lighting, without flash, with a white paper positioned next to the eye to standardize white.

Recently, some authors have applied more sophisticated techniques to correlate the color of the conjunctiva with the level of hemoglobin. In [44], the K-means algorithm is used to group the homogeneous color pixels extracted from the conjunctiva images, represented in RGB format. The authors argue that using this technique can greatly increase the accuracy of anemia estimation.

In [45], the hue saturation intensity color space is used to represent the shades of the conjunctiva image acquired via smartphone. In addition, the use of a previously trained Elman Neural Network allows us to classify anemic and non-anemic patients effectively. The saturation and hue of the pixels are used as features in this study.

*4.2. Estimation of Anemia Through Fingertip Analysis*

Many authors have also explored the use of a smartphone for the accurate monitor of hemoglobin concentration recording and analyzing the color signals of a fingertip placed in contact with its camera. Some methods are based on the use of external LED or additional sensing devices, while studies have been reported that analyze the fingertip image data with alternative approaches to mitigate those issues.

In [46], HemaApp is presented, a smartphone application that monitors the concentration of hemoglobin in the blood in a non-invasive way using the smartphone's camera and special light sources that illuminate the patient's fingertip. The study focuses on three methods based on different acquisition configurations, namely with the white flash + infrared emitter, with the white flash + infrared emitter + incandescent lamp and, finally, with the white flash + series of infrared lights. In [47] the same authors improve the hardware configuration by amplifying the weaker signals of green and blue. In this way, a better color balance allows the simultaneous measurement of different wavelengths.

Further, in [48] the research concerning the comparison between clinical measurements of Hb with the information obtained from a video of the fingertip is presented. The smartphone camera was used. The dataset considered is very small, consisting of only five patients with sickle cell anemia, but this number seems significant for the study carried out. The patient sample considered has been considerably expanded and a new experiment has been presented by the same authors in [49]: people from different areas of the world have been taken into consideration to try to outline a general model that allows to better estimate the anemia. The same authors, in [50] present a prediction model of the

hemoglobin level which demonstrates an accuracy of 95%. In this research work, RGB information extracted from a fingertip video, converted into different color spaces, and considering the hue, saturation, value, brightness, a*, b* and gray components were used. The features are built with all possible combinations of ten different colors and a regression algorithm based on the partial least squares (PLS) method is applied. In addition, the best color combination is evaluated to generate a prediction model with data collected from patients. Furthermore, in [51] the same authors have made improvements to the previous work by analyzing a greater number of frames in which the red, green and blue pixel intensities were estimated taking into consideration 'blocks' (the image is divided into multiple regions, blocks, which contain the average pixel intensity). Basing on blocks on 300 frames an artificial neural network was used in order to predict the level of hemoglobin.

### 4.3. Estimate of Anemia by Analyzing the Nail Bed

In the literature, there are few studies on the analysis of the nail bed to estimate the level of hemoglobin concentration in the blood and, specifically, to hypothesize the condition of anemia in a subject. Even we ourselves do not place much trust in the analysis of the nail bed in accordance with some preliminary experiments that we are carrying out.

In an interesting study by Mannino et al. [52], it is claimed that the nail bed is an ideal area to consider, as it has minimal amounts of melanin compared to other parts of the body, making this methodology independent of the ethnic groups considered for screening.

In this study, the authors extract the color features from the nail bed images converted into the CIE lab color space. In particular, the L value is shown to be a good indicator of skin tone. The author claims that there is a relationship between the hue of the skin and the Hb values, by measuring the value of feature L relative to a flap of skin adjacent to the nail. However, it appears that the measurement is not very reliable. The authors themselves recognize that the observation of this tissue may be influenced by diseases that cause discoloration of the nail bed, such as jaundice and cyanosis.

## 5. Summary of the Studies

As it has been shown in the previous sections, the efforts made by several international research groups are remarkable, and here we have reported only a small number. The goal of diagnosing a serious and unpleasant pathology such as anemia with technologies for personal use is commendable.

We wanted to carry out a more in-depth analysis of the different ways of presenting the published works. We do not pretend to be exhaustive and in many cases we expose our partisan judgment. But we are convinced that this effort will stimulate our friends and fellow researchers also to find a common work agenda. It must be said that in some cases we have found significant differences in the presentation of the results obtained from the experiments carried out and this makes the synergic comparison of the researchers difficult.

In the published papers that we found in the literature, we looked for characteristics of work organization useful to compare the results obtained, with the aim of aggregating elements and work ideas and new research directions for each of us. From the tables the reader takes away the data at a glance: the data summarized in the tables are placed at the basis of our considerations in the final discussion section.

Table 1 highlights:

– The correlation that the authors found between the observation of the tissues and the level of hemoglobin concentration in the blood;
– If the authors have tried to identify the cases of patients suffering from anemia through a classification and therefore what type of classifiers are indicated;
– The number of false negatives found. In addition to the sensitivity/specificity measures, we consider it interesting to understand how many cases can occur in which the 'alarm' is not set;
– Classifier performance indicators (accuracy, specificity, sensitivity, precision).

**Table 1.** Characteristics of the experiments carried out by the authors of the papers taken into consideration here. Keys: SVM (support vector machine), ANN (artificial neural network), LR (linear regression), DT (decision tree), k-NN (k nearest neighbors), LS-SVM (least squares support vector machine), LR (linear regression), R (regression), MLR (multivariate linear regression), kM (k-Means).

| Paper | Correl. Index (Hb) | Classifier | # False Negatives | Accuracy | Specificity | Sensibility | Precision |
|---|---|---|---|---|---|---|---|
| Suner2007 [32] | 0.6 | - | 15 | 0.71 * | 0.72 | 0.69 | 0.64 * |
| Bevilacqua2016 [33] | 0.49 | SVM | 0 | 0.84 | 0.82 | 1 | 0.43 * |
| Irum2016 [34] | - | LS-SVM | 7 | 0.85 | 0.7 | 0.92 | 0.92 * |
| Chen2016 [35]–Q = quick Alg./R = robust Alg. | - | SVM/ANN | 8 (Q)/9 (SVM)-10 (ANN) (R) | 0.32 *(Q)/ 0.81 *-0.80 * (R) | 0.90(Q)/ 0.83-0.83 (R) | 0.62(Q)/ 0.78-0.75 (R) | 0.81 *(Q)/ 0.76 *-0.75 * (R) |
| Chen2017 [36] | - | R | - | - | - | - | - |
| Anggraeni2017 [43] | 0.92 | LR | - | - | - | - | - |
| Wang2016 [46] | 0.69/0.74/0.82 | SVM | - | - | 0.71/0.71/0.77 | 0.79/0.86/0.86 | - |
| Wang2017 [47] | 0.62 | LR | - | - | - | - | - |
| Tamir2017 [42] | - | - | 4 | 0.79 | - | - | - |
| Hasan2017 [48] | - | LR | 0 | - | - | - | - |
| Ahsan2017B [49] | 0.56 | DT/SVM/k-NN | - | 0.49 | - | - | - |
| Hasan2018 [50] | - | MLR/DT | - | 0.95 | - | - | - |
| Hasan2018B [51] | 0.93 | ANN | - | 0.92 | 0.96 | 0.94 | - |
| Dimauro2018A [37] | 0.75 | k-NN | 0 | 1 * | 0.4 * | 1 * | 1 * |
| Dimauro2018B [38] | 0.74 | - | - | - | - | - | - |
| Sevani2018 [44] | - | kM | - | 0.9 | - | - | - |
| Muthalagu2018 [45] | - | ANN | 5 | - | 0.96 | 0.77 | - |
| Mannino2018 [52] | 0.26 | - | - | 0.95 | 0.76 | 0.92 | - |

(*) values calculated by the authors of this paper.

Equally interesting, it seemed to us to extract and group in Table 2 the characteristics of the datasets described by the authors, together with the thresholds taken into consideration to define, in the datasets themselves, the anemic/non-anemic class of the persons considered.

**Table 2.** Characteristics of the datasets and thresholds. Key: SA (severe anemia), MA (mild anemia), mA (moderate anemia), C (conjunctiva), F (fingertip), N (nailbed), M (male), F (female), P (pregnant).

| Paper | Samples | Ethnicity/ Nationality | Age | Anemic | Gender | Hb Threshold (g/dL) | Tissue |
|---|---|---|---|---|---|---|---|
| Suner2007 | 63 | Various ethnicities | 17–95 | - | 41 M/22 F | Hb < 11 | C |
| Bevilacqua2016 | 77 | Italy | - | 9 | - | SA: Hb < 10.5 MA: 10.5 < Hb < 11.5 | C |
| Irum2016 | - | Pakistan | - | - | - | SA: 6 < Hb < 7.5 MA: 8 < Hb < 9 | C |
| Chen2016 | 100 | - | - | - | - | Hb < 11 | C |
| Chen2017 | 100 | - | - | 40 | - | Hb < 11 | C |
| Anggraeni2017 | 20 | Indonesia | 22–36 | - | 20 P | MA: 10 < Hb <10.9 mA: 7 < Hb < 9.9 SA: Hb < 7 | C |
| Wang2016 | 31 | Various ethnicities | 6–77 | 14 | 15 M/16 F | - | F |
| Wang2017 | 32 | - | 18–35 | - | - | - | F |
| Tamir2017 | 19 | Bangladesh | - | 10 | 7 M/12 F | - | C |
| Hasan2017 | 30 | U.S.A. | - | 5 | - | Hb < 10 | F |
| Hasan2017B | 84 | Bangladesh | - | - | - | SA: Hb < 10 MA: 10 < Hb < 11.5 | F |
| | 17 | U.S.A. | - | - | - | | |
| | 49 | U.S.A. | - | 29 | - | | |
| Hasan2018 | 5 | U.S.A. | - | 5 | - | - | F |
| | 75 | Bangladesh | 20–56 | - | 20 M/55 F | | |
| Hasan2018B | 75 | Bangladesh | 20–56 | 39 | 20 M/55 F | Hb < 10.8 | F |
| Dimauro2018A | 113 | Italy | - | - | 63 M/50 F | SA: Hb < 10.5 MA: 10.5 < Hb < 11.5 | C |
| Dimauro2018B | 65 | Italy | 20–80 | 12 | 47 M/18 F | | |
| Sevani2018 | 10 | - | - | - | 6 M/4 F | - | C |
| Muthalagu2018 | 127 | India | - | 22 | - | Hb < 10 | C |
| Mannino2018 | 50 | U.S.A. | 1–62 | - | - | Hb < 12.5 | N |

## 6. Discussion and Conclusions

The main purpose of this work was not to produce a new review: in fact, there are already works of this type of great value made by research friends in different countries of the world. We counted more than one hundred papers that, directly or indirectly, could be considered, and these numbers demonstrate how relevant the ferment in this sector is that unites computer science, bio-engineering and medicine. But the sample of papers taken into consideration was sufficient to achieve the aim set by our study: understand what the open problems are, what critical issues appear insurmountable, what common mistakes we are making, if there is the possibility of defining a road-map and a common agenda. In this sense, we want to highlight here some issues that seem worthy of common discussion and the subject of synergistic agreements. From what we write now it will be understood that it is important to collaborate on common ground, obviously leaving freedom to follow new paths, verify the goodness of new ideas and experiment with new methods.

### 6.1. Some Preparatory Issues

We focus here on the conjunctiva, which finds the greatest number of findings in the literature. Careful study and knowledge of the anatomy of the conjunctiva, in particular of the eyelid, are behind the development of a solution to support the diagnosis of anemia. The mucous membrane extends from the inner palpebral margins up to the eyeball, covering the arches and excluding the corneal area. We can say that the high degree of vascularization, guaranteed by the abundant presence of micro-vessels, makes the conjunctiva a perfect candidate for observation during the physical examination. The palpebral conjunctiva, compared to the forniceal conjunctiva, better highlights the vascularization of the underlying area and probably allows one to highlight the minimum variations in blood color. The assumption seems confirmed by the scientific literature. However, some authors take into consideration the whole conjunctiva, palpebral and forniceal, for the construction and validation of their models. We should strive to clarify this point; it is an open problem.

Another problem, apparently trivial, is to establish whether we can say that the investigations carried out on a small portion of the conjunctiva can be sufficient to affirm that the result is independent of the position in which the portion of the eyelid is carried out. In fact, the sparsity and density of the blood micro-vessels can change in different parts of the eyelid. Often, in some parts of the selections that we use there is no presence of micro-vessels, can we argue about this problem?

Very often the eye diseases are bilateral, but in many cases also unilateral. How many digital images, of one eye or both eyes, are sufficient for a reliable evaluation? Perhaps it is enough to analyze only one eye and certainly little difference makes whether to use the right or the left. New experiments in this regard are welcome. But even more important is to define a common protocol that recommends the cases in which to avoid the analysis through the systems discussed in this paper. For which diseases is it not recommended? Only in case of specific ocular pathologies or, as some authors claim, should other systemic conditions also lead to exclusion, such as abnormal bilirubin values?

We trust that these tools will have an interesting perspective both for the monitoring of the pathology and also for early detection: who has well understood what a big problem is anemia immediately understands that both these opportunities are of great interest.

In support of home monitoring and to guarantee patient health, the sharing of 'withdrawals' should be guaranteed by Electronic Medical Records (EMR) platforms, which have been discussed since time immemorial and which should already be available. The personal physician or any care provider chosen by the patient should be able to remotely assess the patient's condition in general, but referring here specifically to the assessment of the state of anemia. Therefore, each instrument should be equipped with a system for remote verification. Early detection can instead be facilitated by the availability of the anamnesis, even a synthetic one. Even only the age and gender of the patient can make the difference, given the distribution of the disease in the population. In addition, in this sense, EMR platforms could be important.

Two other questions should be crucial in this sense: can the simultaneous observation of multiple tissues (i.e., conjunctiva, fingertip and nailbed) also contribute to determining a more reliable estimate?

Furthermore, it may be more useful to concentrate on the generalization of the instruments or to focus on the personalization of the estimate: it is foreseeable that methods and tools that can be easily calibrated on the individual patient can guarantee a decidedly superior reliability. Therefore, it is important to define parametric models that can quickly be customized, even by the patient himself, or, at most, by his doctor. It is essential to remember here that the tools we are discussing must be reliable and should be usable even directly by the patient, when he is capable, in his own home.

### 6.2. More Specific Technical Issues

When focusing on some more technical aspects, one of the first questions we asked ourselves, but also many other authors, was whether we could freely photograph the eye area at close range with the certainty of not causing damage to the patient. There are demonstrations in the literature that the LED lights used by smartphones cannot in any way damage the delicate human visual system.

Here, experiments of this type, fundamental for the entire work that we are all doing, must be strongly encouraged and placed at the basis of indications accepted by the certification institutions of all the interested nations.

We observed in the previous paragraphs that the interest of the researchers is directed both to the estimate of the hemoglobin concentration and to the determination, perhaps a simpler task, of the anemic condition. Even simpler is probably the determination of the state of severe anemia. In fact, many results have been presented on the estimate of hemoglobin concentration: very interesting correlation indices have often been presented, but we know that the correlation provides a general indication, stimulates us to deepen certain ideas, but is often a double-edged sword, as we would say of the average. Greater confidence inspires methods that target the determination of severe anemia. Perhaps the reliable estimate of the level of hemoglobin concentration in the blood by observing the exposed human tissues is too ambitious? We'll see.

Two other more niche themes deserve some attention. When we acquire digital images using a smartphone, we will be able to evaluate pallor through RGB data. We will be able to study any transformation in any color model, use components of models that are more inspired by the human visual system or that are more 'linear' with the variation of the color, red in particular. But we will all start from the same point, i.e., pixels acquired with the sensors that mass device manufacturers design for them, RGB data and a more or less complex mathematical transformation. In addition, again, are RAW or Jpeg/PNG images equivalent for our purposes?

We noticed a lot of effort by the authors to define automatic ROI segmentation techniques. From the most classic Otsu and watershed algorithms to the more complex and recent methods based on deep learning. Further, then there are intermediate solutions with assisted manual selection. In fact, here too, in our opinion, we can find useful and definitive solutions, but these will only be definitive when we have established with certainty which part and how much of the conjunctiva to consider, as we explained above.

*6.3. Data Sharing and Threshold Assessment*

Researchers around the world are continuously working with data. The major success cases of international research are typically obtained through coordinated work. Medicine is an excellent example. Another excellent example is data sharing for famous cases concerning epidemics. Many publishers provide data platform that supports open data initiatives and gives everyone the opportunity to manage, share, access and store research data. We should agree and share our data, ensuring patient privacy and safety. Each of us will have the opportunity to freely access all our shared datasets and be able to analyze and use them with proper citation. Sharing data would also become very useful for dealing with problems that in some cases afflict medical research, such as the imbalance of datasets. An imbalanced learning problem appears in the supervised classification when one of the classes is not sufficiently represented. We have noted that class imbalance condition affects in some cases the topic here presented. Many papers have focused on the implications of ignoring this problem, as well as designing suitable solutions to solve or relieve it. The presence of a marked imbalance in the class distribution may lead to consequences in obtaining a reliable classification task: simply, rare examples are ignored and classification rules can accommodate to the prevalent class. But particular usefulness could have the experimentation of our methods and devices with data coming from different countries and belonging to people of different ethnicity. Without underestimating the fact that the comparison and validation of our methods and techniques becomes much easier and more useful.

Epidemiological estimates of anemia are subject to variations depending on the definition of the limit parameters and the quality of the analytical samples used. Given the wide scope of the problem, epidemiological studies in recent decades are partial and often questionable. The World Health Organization carried out empirical studies that find applicative feedback, but it must be noted that some authors dispute the above studies. However, there is an open debate concerning which hemoglobin lower threshold should be considered to define anemia and consequently how can we treat the data that

we possess. Recently, new lower limits have been proposed based on the databases NHANES-III and Scripps-Kaiser, in which the hemoglobin concentration was determined with standardized automated methods: these limits are 13.7 g/dL for white men (20–59 years) and 13.2 g/dL for men above the age of 60 years; the corresponding value for women is 12.2 g/dL independently of age. In Afro-Americans, these limits are lower: 12.9 g/dL in younger men and 12.7 g/dL in men older than 60 years, while the corresponding value for women is 11.5 g/dL for all ages. Should we use these thresholds and consequently update our datasets?

## 7. Future Work

Non-invasive tools such as those discussed here are of considerable social importance and the objections raised by those who want to criticize our research are worth little: someone asserts that with 'less invasive' tools that take only a drop of blood you can get good measurements. We talk about different issues, different costs, different personal discomfort and much broader functionality.

We hope that many authors will find interest in setting up an international interest group that discusses specific issues of the topics we have dealt with here, which is interested in defining and disseminating best practices and can act as a reference for dialogue with stakeholders, including smartphone manufacturers or international bodies such as WHO. This can help to propose further guidelines, based on the products and results of our research, in documents such as, by way of example, the guidelines for integrated management of childhood illness.

Finally, we are confident that new software technologies based on machine or deep learning will be able to provide powerful solutions to support our methods and techniques and to some of the open problems highlighted by us. New and increasingly powerful smartphone-based devices will soon be available to us. In essence, technology will certainly play our part, but it remains to be understood whether the issues discussed here are of such characteristics that new technologies will be able to 'automatically' solve or not.

We hope that this paper and in particular the discussion could be the starting point for an open debate about the dissemination of our experiments and pave the way for further updates and improvements of what we have outlined.

**Author Contributions:** Conceptualization, G.D., S.D.R. and A.R.; data curation, F.D.T.; formal analysis, V.V. and F.G.; investigation, V.V. and L.C.; methodology, G.D., A.R. and F.G.; project administration, G.D.; resources, G.D. and F.G.; validation, L.C. and F.D.T.; writing—original draft, G.D. and S.D.R.; writing—review and editing, S.D.R., F.D.T., A.R., V.V. All authors have read and agreed to the published version of the manuscript.

**Funding:** This research received no external funding.

**Conflicts of Interest:** The authors declare no conflict of interest.

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
