# Peer review of "Estimate of Anemia with New Non-Invasive Systems—A Moment of Reflection"

_electronics, doi:10.3390/electronics9050780_

Round 1

Reviewer 1 Report

Thank you for a very informative paper that introduces valuable insight into the diagnosis of anemia. Overall, the paper is well written although there remains some confusion about the rigor of this work. Some specific comments below:

  1. The abstract would benefit from a re-write and a more structured approach. Currently it is not a good reflection of the content of the paper. There is little reference to the background, methods or, most notably, what was found. It almost seems like a mini-discussion. It would be helpful to list the main outcomes from your literature search and provide strong conclusion based on these. The wording "kind of guideline" does not fill me with confidence in your conclusion.
  2. The introductory is excellent and really sets the scene for the paper. There is, however, a lack of references to literature and it would be beneficial to provide a clear aim statement to justify the objectives of your study. 
  3. The next 2 sections don't seem to flow well from the intro. It would be helpful to know what method you utilised to search for your literature and what inclusion/exclusion criteria were used. You mentioned hundreds of studies but how did you select those that are discussed? A methods section would be useful to describe this. 
  4. The descriptions of the diagnostic tools are good and provide a good case for the utilisation of non-invasive techniques. However, this in depth information suggests the aim of your paper is about describing non-invasive techniques rather than reflecting on their use.
  5. Line 264 is not in English (on purpose??). 
  6. Section 5 does not align well within your reflection - is this a critical analysis? What do you want the reader to take away from the tables? Validity? Reliability? Research rigor for each diagnostic tool?  It would be beneficial to structure your paper into sections that clearly define the content.
  7. The discussion is fair, however again the focus of the paper seems to have shifted. You are presenting issues with the diagnostic techniques, is this a critical analysis? Or a reflection?  
  8. The first paragraph of the conclusion is great - if the remainder of the paper aligned to this, it would be much clearer. However, as the conclusion goes on, there is a shift in direction and focus which loses the reader. 

Overall, the paper requires a re-structure and clearer aims and objectives. 

Author Response

ANSWERS TO REVIEWER 1

Thank you for a very informative paper that introduces valuable insight into the diagnosis of anemia. Overall, the paper is well written although there remains some confusion about the rigor of this work. Some specific comments below:

  1. The abstract would benefit from a re-write and a more structured approach. Currently it is not a good reflection of the content of the paper. There is little reference to the background, methods or, most notably, what was found. It almost seems like a mini-discussion. It would be helpful to list the main outcomes from your literature search and provide strong conclusion based on these. The wording "kind of guideline" does not fill me with confidence in your conclusion.

Answer

We thank the reviewer for the efforts he has devoted to providing valuable suggestions to improve our paper and for the gratifying comments he has expressed in some cases.

We specified in the abstract that the main purpose of this work was not to produce a new review. Therefore, our work does not require the methodological rigor necessary in writing a review.

But the reviewer is right. We have explained now better that we wanted to write a reasoned and justified commentary, as we describe our personal experience of this specific topic and give our own opinions and perspectives. We think that anemia estimation through smartphone-based devices and methods is a current, hot and often controversial subject, then we provide our personal views of and insight into this topic.

Our paper does not have the aim to introduce new information, and we limited the number of references as requested in a commentary paper to support the author's opinion.

We apologize for the unstructured abstract, generally commentary requires it, but we are convinced that the reviewer wants to help us improve our paper, so we have completely rewritten the abstract using a slightly more rigorous form.

  1. The introductory is excellent and really sets the scene for the paper. There is, however, a lack of references to literature and it would be beneficial to provide a clear aim statement to justify the objectives of your study.

Answer

The first part of the introduction is intended to guide the reader towards the general problem of anemia. Indeed, we have introduced not so many references, then we have improved this section introducing new references on the medical aspects of anemia and inserting a short description about epidemiology with related references.

In the final part we add clear aim statements to justify the objectives of the following sections.

  1. The next 2 sections don't seem to flow well from the intro. It would be helpful to know what method you utilised to search for your literature and what inclusion/exclusion criteria were used. You mentioned hundreds of studies but how did you select those that are discussed? A methods section would be useful to describe this.

Answer

Section 2 has the main aim to show that it is worthwhile to expand these research works and develop new technologies, with some important purposes: significantly reduce the costs borne by the national health systems, power the healthcare and medical services that would be exempted from a considerable amount of practically useless activities and finally (but not in order of importance) to obtain positive effects on patients, for example, those who must frequently undergo laboratory analysis: this is a value that is difficult to estimate but even more relevant than the previous ones. In section 3 we show some approaches for the clinical evaluation of anemia oriented to the analysis of different exposed tissues; the most promising efforts were dedicated to the measurement of haemoglobin by analysing the colour of the fingertips, nail bed and the palpebral conjunctiva.

We have clarified these objectives better and introduced them at the end of section 1 (Introduction).

As for the second part of the reviewer's question in point 3, we have been studying these topics for some years and actually dozens of articles have been written on this topic. This answer it is useful to better explain both the doubts expressed also in point 4 by the reviewer: since we are not going to write a review, we have limited ourselves to considering some articles that we know well and use them to dissect the problems and critical issues that we intend to discuss. We specified that it was not a selection of works based on quality or importance, because it would have been offensive to many other authors who have written excellent works. But few works have been sufficient to identify important critical issues, on which we want to reason and stimulate discussion with other researchers. Furthermore, as we indicated on the last page of our paper, we intended to include a complete supplementary bibliography which contains, not a selection, but all the articles that we consider interesting on this topic. Of course, if reviewers consider it an unnecessary addition, we can review this choice.

The articles have been selected from the Scopus, Web of Sciences and Google Scholar databases, using a search key basically built on the words 'anemia' and 'noninvasive devices', with some variations and additional keywords. This process will be better defined in a paper that rigorously describes a Systematic Mapping Study that we are completing and that we will shortly submit: but the latter has a different purpose.

We added this last description in the last part of our paper.

  1. The descriptions of the diagnostic tools are good and provide a good case for the utilisation of non-invasive techniques. However, this in depth information suggests the aim of your paper is about describing non-invasive techniques rather than reflecting on their use.

Answer

Thanks to the reviewer's suggestions, I think we have now better explained the purpose of this work also in the other responses and improvements to the paper.

In section 4 we report a brief overview of some technologies to support an effective diagnosis of anemia, focusing on the most promising efforts, based on low-cost and non-invasive devices, such as smartphones. As we have written now in the last part of introduction, there we introduce also some basic methods and principles of operation of some of these devices and highlight some limitations and some obvious technical problems. We wanted to present this section, also to clearly differentiate the most popular approaches at the moment, which concern the analysis of the tissues of the conjunctiva, the fingertip and the nailbed.    

  1. Line 264 is not in English (on purpose??).

Sorry, we missed it. now we have translated it.

  1. Section 5 does not align well within your reflection - is this a critical analysis? What do you want the reader to take away from the tables? Validity? Reliability? Research rigor for each diagnostic tool? It would be beneficial to structure your paper into sections that clearly define the content.

Answer

Thank you, also in this case the reviewer's criticisms help us to be more rigorous and understandable.

Section 5 describes an observation of the facts, a clear demonstration that researchers behave differently in carrying out the experiments and in presenting the results, which in this way become little comparable. The rigor with which the researches are carried out is not evaluated here.

From the tables the reader takes away the data at a glance: the data summarized in the tables are placed at the basis of our considerations in the following sections. In this way, the reader can verify with certainty that what we say is not cleared in the air, but is evident from the data that we have highlighted in that tables.

It can be considered a critical analysis of different behaviors, which concerns all of us researchers in this field.

Thanks again, we wrote these concepts better in section 5.

  1. The discussion is fair, however again the focus of the paper seems to have shifted. You are presenting issues with the diagnostic techniques, is this a critical analysis? Or a reflection?
  2. The first paragraph of the conclusion is great - if the remainder of the paper aligned to this, it would be much clearer. However, as the conclusion goes on, there is a shift in direction and focus which loses the reader.

Answer

Thanks, we think that after explaining our goal in the previous answers and improvements that we made to the paper, the conclusions are now understandable. We hope the reader will not get lost now.

But as the reviewer has perfectly highlighted the critical issues of our paper, we are willing to further improve the discussion if the reviewer deems it useful.

Last point: Overall, the paper requires a re-structure and clearer aims and objectives.

Answer

We think we have answered the reviewer's doubts and made significant improvements to the paper. We still want to praise the auditor once again for the help he has provided.

Reviewer 2 Report

-The paper presents a succinct and useful analysis of the literature review on novel non-invasive systems to estimate anemia.
-The technical aspects of the research presented seem sound and detailed.
-The topic and area of research are relevant.

I have some minor suggestions regarding style and typos in the paper.

-Line 24: I think you should use the terms "studies" or "research works" instead of "researches".
-Line 95: I think it should be "are" instead of "is" in "advances in digital image processing is playing a...".
-Line 114: I think the work "to" should be added between the words "suggest" and "add" in "the authors suggest add further signs...".
-Line 191: In this case it might be "Institute" instead of "Institutes".
-Line 253: I think it should be "mostly" instead of "most" in "which occurs most in Africa and Asia.".
-Line 264: The title of this subsection is in Italian.
-Line 268: I think it should be "approaches" instead of "approach" in "the fingertip image data with alternative approach to...".
-Line 274: There is a space between the word "lights" and the full stop at the end of the line.
-Line 326: I think it should be "Table 1" instead of "Tab.1".
-Line 336: I think it should be "Table 2" instead of "Tab.2".
-Line 443: I think it should be "concerning" instead of "concern" in "sharing for famous cases concern epidemics.".
-Line 447: I think it should be "be" instead of "are" in "shared datasets and are able to...".
-Line 464: I think it should be "can we" instead of "we can" at the end of the line.
-Line 468: I think it should be "above" instead of "after" in "for men after the age of 60 years;".
-Line 484: I think it should be "machine" instead of "machines" in "technologies based on machines or deep learning...".
-Refer to specific tables or figures putting the first letter in capital, for example: "see Figure 1".

Author Response

ANSWERS TO REVIEWER 2

Comments and Suggestions for Authors

-The paper presents a succinct and useful analysis of the literature review on novel non-invasive systems to estimate anemia.

-The technical aspects of the research presented seem sound and detailed.

-The topic and area of research are relevant.

I have some minor suggestions regarding style and typos in the paper.

-Line 24: I think you should use the terms "studies" or "research works" instead of "researches".

Thanks, fixed also in other lines.

-Line 95: I think it should be "are" instead of "is" in "advances in digital image processing is playing a...".

Thanks, fixed.

-Line 114: I think the work "to" should be added between the words "suggest" and "add" in "the authors suggest add further signs...".

Thanks, fixed.

-Line 191: In this case it might be "Institute" instead of "Institutes".

Yes, correct, fixed, thank you.

-Line 253: I think it should be "mostly" instead of "most" in "which occurs most in Africa and Asia.".

Yes, right, fixed, thank you.

-Line 264: The title of this subsection is in Italian.

Sorry, we missed it. now we have translated it.

-Line 268: I think it should be "approaches" instead of "approach" in "the fingertip image data with alternative approach to...".

Yes, right, fixed, thank you.

-Line 274: There is a space between the word "lights" and the full stop at the end of the line.

Fixed, thank you.

-Line 326: I think it should be "Table 1" instead of "Tab.1".

-Line 336: I think it should be "Table 2" instead of "Tab.2".

Fixed both, thank you.

-Line 443: I think it should be "concerning" instead of "concern" in "sharing for famous cases concern epidemics.".

Yes, right, fixed, thank you.

-Line 447: I think it should be "be" instead of "are" in "shared datasets and are able to...".

Yes, right, fixed, thank you.

-Line 464: I think it should be "can we" instead of "we can" at the end of the line.

Yes, right, fixed, thank you.

-Line 468: I think it should be "above" instead of "after" in "for men after the age of 60 years;".

Yes, better, fixed, thank you.

-Line 484: I think it should be "machine" instead of "machines" in "technologies based on machines or deep learning...".

Of course, fixed, thank you.

-Refer to specific tables or figures putting the first letter in capital, for example: "see Figure 1".

Done, thank you.

Round 2

Reviewer 1 Report

Thank you to the authors for a timely, thorough response to the review comments. Overall, the paper is very much improved. There are some minor comments below. One major comment I will make is that the aim of your paper is not aligned with what is stated. After reading this paper again, I feel that theaim is more along the lines of "The aim of this paper is to review current literature to ascertain areas for further research in the diagnosis of anaemia" This fits much better with the overall sections. 

Another major comment in that you state this is a review of studies, however you spend a lot of time describing the actual diagnostic tools, rather than reviewing them. It would be beneficial to reduce Section 4 a lot and add further data to Section 5 - your review. 

  1. The abstract is much clearer and provided improved information about what the paper is aiming to do. However, there are some areas that could be further improved. The background section contains information about the procedure "this paper deals with...." (Line 16-18) This sentence can be moved or removed. Line19-21 is also not appropriate for the background. This belongs in the findings. The conclusions states that "It will emerge...." (Line 34-35) this sentence is inappropriate and can be removed. Similarly, Lines 35-37 should not start with "we think" - just state what the findings have told you (for example "This paper could be the starting point..."
  2.  The references in your introduction has improved. Well done. 
  3. Line 78 - needs a reference (many studies ?)
  4. Line 89 - what does a "hot" subject mean? I think this term is irrelevant
  5. I would like to see Line 88 (this paragraph) start with "The aim of this paper is to provide a commentary to highlight...." Currently this whole paragraph is poorly written. 
  6.  Line 92 should start "The aim of Section 2 is..."
  7.  Line 92 - 104 - I think this section is confusing and does not give a clear overview of the paper. If you amend Lines 88-91 and make the aims and objectives clear, you can remove this paragraph all together. 
  8.  Lines 135 - 140 are poorly written. Line 135 could read "Expanding these current research and the development of new technology may significantly reduce the costs....etc"
  9. Line 141 - Try not to say "we will see". Use terms like We explore or we provide evidence for.... 
  10.  Section 5 - Is this s summary of studies? Or of techniques? Are you only discussing 1 study per section? I would re-name your section title to better reflect what you are going to describe. 
  11. Line 371 - Try not we say "we tried" - it is better to state what you have done. e.g. Table 1 highlights...
  12.  Lines 395 - 398 are inappropriate. Either word them better or remove. 
  13. Your conclusion still needs work - try not to say "we talk about", use language that summarises your findings. 

Author Response

Answer to reviewer 1

Thank you to the authors for a timely, thorough response to the review comments. Overall, the paper is very much improved. There are some minor comments below.

Thank you, many improvements are due to the valuable suggestions of the reviewer. Now we will also follow these new suggestions

One major comment I will make is that the aim of your paper is not aligned with what is stated. After reading this paper again, I feel that the aim is more along the lines of "The aim of this paper is to review current literature to ascertain areas for further research in the diagnosis of anaemia" This fits much better with the overall sections. Another major comment in that you state this is a review of studies, however you spend a lot of time describing the actual diagnostic tools, rather than reviewing them. It would be beneficial to reduce Section 4 a lot and add further data to Section 5 - your review.

Thanks for the comments. As we have specified several times in our paper, this work is not a review (please see: “the main purpose of this work was not to produce a review” (abstract), “Since we are not going to write a review” (section 4), “The main purpose of this work was not to produce a new review” (section 6), “The aim of this paper is to provide a commentary” (introduction)); similarly, we have also replied to the kind reviewer in the first round. We have written a commentary paper that has a very different purpose and follows less strict rules than a review. However, we were honored to adhere to the auditor's requests to outline a better and more rigorous framework for our paper. We believe that it is now ready for publication, and it is fair to recognize that this result has been achieved thanks to the meticulous work of the reviewer, whom we thank again.

The abstract is much clearer and provided improved information about what the paper is aiming to do. However, there are some areas that could be further improved.

- The background section contains information about the procedure "this paper deals with...." (Line 16-18) This sentence can be moved or removed.

- Line19-21 is also not appropriate for the background. This belongs in the findings.

- The conclusions states that "It will emerge...." (Line 34-35) this sentence is inappropriate and can be removed.

- Similarly, Lines 35-37 should not start with "we think" - just state what the findings have told you (for example "This paper could be the starting point..."

Thank you, the abstract has been much improved by strictly following all the above reviewer's suggestions.

The references in your introduction has improved. Well done.

Thank you.

Line 78 - needs a reference (many studies?)

Added some reference, thank you.

Line 89 - what does a "hot" subject mean? I think this term is irrelevant

I would like to see Line 88 (this paragraph) start with "The aim of this paper is to provide a commentary to highlight...." Currently this whole paragraph is poorly written.

Line 92 should start "The aim of Section 2 is..."

Line 92 - 104 - I think this section is confusing and does not give a clear overview of the paper. If you amend Lines 88-91 and make the aims and objectives clear, you can remove this paragraph all together.

Many thanks to the reviewer for these suggestions. We reworded completely this section following the reviewer's directions and now it does give a clear overview of the paper.

Lines 135 - 140 are poorly written. Line 135 could read "Expanding these current research and the development of new technology may significantly reduce the costs....etc"

Much better now, thank you.

Line 141 - Try not to say "we will see". Use terms like We explore or we provide evidence for....

We reworded this section, much better now, thank you.

Section 5 - Is this s summary of studies? Or of techniques? Are you only discussing 1 study per section? I would re-name your section title to better reflect what you are going to describe.

Re-named, thank you.

Line 371 - Try not we say "we tried" - it is better to state what you have done. e.g. Table 1 highlights...

Right! done, thank you.

 Lines 395 - 398 are inappropriate. Either word them better or remove.

The reviewer is right; we have removed them.

Your conclusion still needs work - try not to say "we talk about", use language that summarises your findings.

We reworded several sections, much better now, thank you.